# Isoginkgetin and Madrasin are poor splicing inhibitors

**Michael Tellier** [ID]¤*, **Gilbert Ansa, Shona Murphy**\*

Sir William Dunn School of Pathology, University of Oxford, Oxford, United Kingdom

¤ Current address: Department of Molecular and Cell Biology, University of Leicester, Leicester, United Kingdom

\* mt477@leicester.ac.uk (MT); shona.murphy@path.ox.ac.uk (SM)

## Abstract

The production of eukaryotic mRNAs requires transcription by RNA polymerase (pol) II and co-transcriptional processing, including capping, splicing, and cleavage and polyadenylation. Pol II can positively affect co-transcriptional processing through interaction of factors with its carboxyl terminal domain (CTD), comprising 52 repeats of the heptapeptide Tyr1-Ser2-Pro3-Thr4-Ser5-Pro6-Ser7, and pol II elongation rate can regulate splicing. Splicing, in turn, can also affect transcriptional activity and transcription elongation defects are caused by some splicing inhibitors. Multiple small molecule inhibitors of splicing are now available, some of which specifically target SF3B1, a U2 snRNP component. SF3B1 inhibition results in a general downregulation of transcription elongation, including premature termination of transcription caused by increased use of intronic poly(A) sites. Here, we have investigated the effect of Madrasin and Isoginkgetin, two non-SF3B1 splicing inhibitors, on splicing and transcription. Surprisingly, we found that both Madrasin and Isoginkgetin affect transcription before any effect on splicing, indicating that their effect on pre-mRNA splicing is likely to be indirect. Both small molecules promote a general downregulation of transcription. Based on these and other published results, we conclude that these two small molecules should not be considered as primarily pre-mRNA splicing inhibitors.

## Introduction

The production of a mature mRNA from an eukaryotic protein-coding gene requires both transcription by RNA polymerase (pol) II and co-transcriptional processes, which include RNA capping, splicing, and cleavage and polyadenylation [1]. Recent research has highlighted how transcription and pre-mRNA splicing can influence each other [2–8]. Coupling between transcription and pre-mRNA splicing is thought to occur, at least partially, via the carboxyl-terminal domain (CTD) of the large subunit of pol II, which is composed in humans of 52 repeats of the heptapeptide Tyr1-Ser2-Pro3-Thr4-Ser5-Pro6-Ser7. The pol II CTD can be modified by several post-translational modifications, including phosphorylation of Tyr1, Ser2, Thr4, Ser5, and Ser7 [9]. While Ser2-P is associated with transcription elongation and Ser5P with transcription initiation and pol II pausing [10], Ser2-P and Ser5-P have also been found

**Data Availability Statement:** All mNET-seq files are available from the Gene Expression Omnibus database (accession number GSE221279, https://www.ncbi.nlm.nih.gov/geo/query/acc.cgi?acc=GSE221279). The raw images of the western blots

and agarose gels are available on Zenodo (https://zenodo.org/doi/10.5281/zenodo.13694860).

**Funding:** This work was supported by the Wellcome Trust [WT106134AIA and WT210641/Z/18/Z to Prof. Shona Murphy]. The funders had no role in study design, data collection and analysis, decision to publish, or preparation of the manuscript.

**Competing interests:** The authors have declared that no competing interests exist.

to specifically interact with splicing factors [11, 12], providing a physical link between the pol II complex and the spliceosome. A direct interaction between the pol II complex and the U1 snRNP was also observed by cryo-EM [13]. In addition, the pol II elongation rate, which can be modified by cellular stresses or by mutating pol II, can affect the inclusion of exons in or the exclusion of exons from the mature mRNA in yeasts, mammals, and plants [2–4, 6, 14–18]. In turn, inhibition of pre-mRNA splicing by Spliceostatin A (SSA) or Pladienolide B (PlaB), two small molecule inhibitors of the U2 snRNP protein SF3B1, affect pol II transcription [5, 7, 19, 20], pol II CTD phosphorylation [5], and recruitment of the key transcription elongation kinase, P-TEFb [7, 21], which is composed of cyclin-dependent kinase (CDK)9 and Cyclin T1.

Coupling between pre-mRNA splicing and cleavage and polyadenylation (CPA) is also required to define the last exon of protein-coding genes and to promote mRNA CPA and transcription termination [1]. This coupling involves interactions between splicing factors, including U2AF65 and the SF3B proteins, and CPA factors, including CPSF proteins and the poly(A) polymerase [22–25]. A consequence of this coupling is that failure to splice the terminal intron in pre-mRNA is associated with a transcription termination defect, as the poly(A) site is no longer recognised by the mRNA CPA complex [19, 26, 27].

To further investigate the effects of pre-mRNA splicing inhibition on transcription elongation and transcription termination, we used two small molecules thought to be inhibitors of the early spliceosome, Madrasin [28] and Isoginkgetin [29]. We found that in contrast to the SF3B1 inhibitors, PlaB and Herboxidiene (HB), Madrasin and Isoginkgetin are poor splicing inhibitors. However, treatment with Madrasin and Isoginkgetin globally downregulates pol II transcription, which was previously observed for Isoginkgetin [30]. These two small molecules should therefore not be considered to be primarily pre-mRNA splicing inhibitors.

## Materials and methods

### Cell culture

HeLa cells were obtained from ATCC (ATCC® CCL-2™) and grown in DMEM medium supplemented with 10% foetal bovine serum, 100 U/ml penicillin, 100 μg/ml streptomycin, 2 mM L-glutamine at 37˚C and 5% $CO_2$. HeLa cells were treated with 30–90 μM Madrasin (Sigma-Aldrich), 30 μM Isoginkgetin (Merck), 1 μM Herboxidiene (Cayman Chemical), 1 μM Pladienolide B (Cambridge Bioscience), or 100 μM 5,6-dichlorobenzimidazone-1-β-D-ribofuranoside (DRB, Sigma) for the time indicated in the figures. As a negative control, HeLa cells were treated with DMSO. Cells were routinely checked for mycoplasma contamination using Plasmo Test Mycoplasma Detection Kit (InvivoGen, rep-pt1).

### Protein extraction and western blotting

For cytoplasmic fractions, HeLa cells were washed twice with ice-cold PBS then scraped into 15 ml tubes and centrifuged at 1,000 rpm for 5 min at 4˚C. The pellet was resuspended thoroughly in 1 ml of Lysis buffer B (10 mM Tris-HCl (pH 8–8.4), 140 mM NaCl 1.5 mM MgCl2, and 0.5% NP40) supplemented with protease inhibitor cocktail (Roche) and PhosSTOP (Roche) by slow pipetting and then transferred to a 1.7 ml tube. The lysate was centrifuged at 1,000 g for 3 min at 4˚C then 500 μl of the supernatant of the unpurified cytoplasmic fraction was transferred to a new tube. Purified cytoplasmic fraction was achieved by centrifugation at 10,000 g for 1 min at 4˚C.

For nucleoplasm and chromatin fractions, the rest of the unpurified cytoplasmic fraction was discarded and the nuclear pellet resuspended in 1 ml of Lysis buffer B. Following the transfer to a 14 ml round bottom Falcon tube, 100 μl of Detergent stock (3.3% (w/v) Sodium Deoxycholate, 6.6% (v/v) Tween 40) was added drop-by-drop under slow vortexing. After a 5 min

incubation on ice, the suspension was transferred to a new ice-cold 1.7 ml tube and centrifuged a 1,000 g for 3 min at 4˚C. The nuclear pellet was resuspended in 1 ml of Lysis buffer B. Following centrifugation at 1,000 g for 3 min at 4˚C, the nuclear pellets were resuspended by pipetting up and down in 125 µl of NUN1 buffer (20 mM Tris–HCl pH 7.9, 75 mM NaCl, 0.5 mM EDTA and 50% (vol/vol) glycerol) and moved to a new 1.5 ml ice-cold microcentrifuge tube. After adding 1.2 ml of ice-cold NUN2 buffer (20 mM HEPES-KOH pH 7.6, 300 mM NaCl, 0.2 mM EDTA, 7.5 mM MgCl2, 1% (vol/vol) NP-40 and 1 M urea), the tubes were vortexed at maximum speed for 10 s and incubated on ice for 15 min with a vortexing step of 10 s every 3 min. The samples were centrifuged at 16,000 g for 10 min at 4˚C and the supernatant kept as the nucleoplasm fraction while the chromatin pellets were washed with 500 µl of ice-cold PBS and then with 100 µl of ice-cold nuclease-free water. The chromatin pellet was then digested in 100 µl of nuclease-free water supplemented with 1 µl of Benzonase (25–29 units, Merck Millipore) for 15 min at 37˚C in a thermomixer at 1,400 rpm.

For chromatin extraction only, a ~ 80% confluent 15 cm dish was washed twice with ice-cold PBS and scrapped in 5 ml of ice-cold PBS. After pelleting the cells at 420 g for 5 min at 4˚C and removing the supernatant, the cells were resuspended in 4 ml of ice-cold HLB + N buffer (10 mM Tris–HCl (pH 7.5), 10 mM NaCl, 2.5 mM MgCl2 and 0.5% (vol/vol) NP-40) and incubated on ice for 5 min. The cell pellets were then underlayed with 1 ml of ice-cold HLB + NS buffer (10 mM Tris–HCl (pH 7.5), 10 mM NaCl, 2.5 mM MgCl2, 0.5% (vol/vol) NP-40 and 10% (wt/vol) sucrose). Following centrifugation at 420 g for 5 min at 4˚C, the nuclear pellets were resuspended by pipetting up and down in 125 µl of NUN1 buffer and transferred into a new 1.5 ml ice-cold microcentrifuge tube. Following the addition of 1.2 ml of ice-cold NUN2 buffer, the tube was vortexed at maximum speed for 10 s and incubated on ice for 15 min with a vortexing step of 10 s every 3 min. The sample was centrifuged at 16,000 g for 10 min at 4˚C and the supernatant discarded while the chromatin pellet was washed with 500 µl of ice-cold PBS and then with 100 µl of ice-cold water. The chromatin pellet was then digested in 100 µl of water supplemented with 1 µl of Benzonase (25–29 units, Merck Millipore) for 15 min at 37˚C in a thermomixer at 1,400 rpm. For loading, 10 µg of proteins were boiled in 1× LDS plus 100 mM DTT.

For whole-cell extract, cells were washed in ice-cold PBS twice, collected in ice-cold PBS with an 800 g centrifugation for 5 min at 4˚C. The pellets were resuspended in RIPA buffer supplemented with protease inhibitor cocktail and PhosSTOP, kept on ice for 30 min with a vortexing step every 10 min. After centrifugation at 14,000 g for 15 min at 4˚C, the supernatants were kept and quantified with the Bradford method. For loading, 20 µg of proteins were boiled in 1× LDS plus 100 mM DTT.

Western blots were performed with NuPAGE Novex 4–12% Bis–Tris Protein Gels (Life Technologies) with the following primary antibodies: Rpb1 NTD (D8L4Y) Rabbit mAb (14958S, Cell Signaling Technology), Phospho-Rpb1 CTD (Ser2) (E1Z3G) Rabbit mAb (13499S, Cell Signaling Technology), Phospho-Rpb1 CTD (Ser5) (D9N5I) Rabbit mAb (13523S, Cell Signaling Technology), Rabbit anti-SF3b155/SAP155 (A300-996A, Bethyl Laboratories), Rabbit anti-α-Tubulin (2144S, Cell Signaling), Proteintech SUMO1 Monoclonal antibody (67559-1-IG-20UL, Proteintech Europe), Proteintech SUMO2/3 Polyclonal antibody (11251-1-AP-20UL, Proteintech Europe), and Histone H3 (ab1791, Abcam). Secondary antibodies were purchased from Merck Millipore (Goat Anti-Rabbit IgG Antibody, HRP-conjugate, 12–348, and Goat Anti-Mouse IgG Antibody, HRP conjugate, 12–349), the chemiluminescent substrate (SuperSignal West Pico PLUS) from Thermo Fisher, and the membranes visualized on an iBright FL1000 Imaging System (Thermo Fisher). Ponceau S–Proteomics Grade staining (VWR International) was performed according to the manufacturer's instructions.

Quantifications were performed with Image Studio Lite version 5.2.5. For each antibody, the quantification has been calculated as: (IP treated signal / histone H3 treated signal) / (IP DMSO signal / histone H3 DMSO signal).

## Chromatin immunoprecipitation followed by qPCR (ChIP-qPCR)

ChIP was performed as previously described [31]. HeLa cells were grown in 100- or 150-mm dishes until they reached ~80% confluence. The cells were fixed with 1% formaldehyde at room temperature for 10 min with shaking. Quenching was performed with 125 mM Glycine for 5 min at room temperature with shaking. Cells were washed twice with ice-cold PBS and then scrapped in ice-cold PBS and pelleted for 10 min at 1,500 rpm for 10 min at 4°C. Lysis was performed on ice for 10 min with ChIP lysis buffer (10 mM Tris–HCl ph8.0, 0.25% Triton X-100, 10 mM EDTA, and protease inhibitor cocktail) before being centrifuged at 1,500 g for 5 min at 4°C. Pellets were washed once with ChIP Wash buffer (10 mM Tris–HCl pH8.0, 200 mM NaCl, 1 mM EDTA, and protease inhibitor cocktail). Pellets were then resuspended in ChIP Sonication buffer (10 mM Tris–HCl pH 8.0, 100 mM NaCl, 1 mM EDTA, protease inhibitor cocktail, and phosphatase inhibitor), incubated 10 min on ice, and sonicated for 30 cycles, 30 seconds on/30 seconds off using a Bioruptor Pico (Diagenode). The samples were centrifuged at 13,000 rpm for 15 min at 4°C. Supernatants were transferred to new Eppendorf tubes and pre-cleared with 10 μl of Protein G Dynabeads for 30 min on a rotating wheel at 4°C. For each IP, 60 μg of chromatin was incubated overnight on a rotating wheel at 4°C with the following antibodies: normal rabbit IgG (2729S, Cell Signaling), RNA polymerase II anti-body (NBP2-32080, Novus Biologicals), Phospho-Rpb1 CTD (Ser2) (E1Z3G) Rabbit mAb (13499S, Cell Signaling Technology), Phospho-Rpb1 CTD (Ser5) (D9N5I) Rabbit mAb (13523S, Cell Signaling Technology), CDC73 (A300-170A, Bethyl Laboratories), SUPT5H antibody (A300-868A, Bethyl Laboratories), and CPSF73 (A301-091A, Bethyl Laboratories). Protein G Dynabeads were saturated overnight with RIPA buffer (10 mM Tris–HCl pH8.0, 150 mM NaCl, 1 mM EDTA, 0.1% SDS, 1% Triton X-100, 0.1% sodium deoxycholate) supple-mented with 10 mg/ml of bovine serum albumin on a rotating wheel at 4°C.

The chromatin-antibody mix was incubated for 1h on a rotating wheel at 4°C with pre-blocked Protein G Dynabeads. IgG supernatant was kept as input. Beads were then washed three times with ice-cold RIPA buffer, three times with ice-cold High Salt buffer (10 mM Tris–HCl pH8.0, 500 mM NaCl, 1 mM EDTA, 0.1% SDS, 1% Triton X-100, 0.1% sodium deoxycholate), twice with LiCl Wash buffer (10 mM Tris–HCl pH8.0, 250 mM LiCl, 1 mM EDTA, 1% NP-40, 1% sodium deoxycholate), and twice with TE buffer (10 mM Tris–HCl pH 7.5, 1 mM EDTA). The samples were eluted twice with Elution buffer (100 mM NaHCO3, 1% SDS, 10 mM DTT) from the beads on a thermomixer for 15 min at 25°C at 1,400 rpm. Each input was diluted 1/10 in Elution buffer. The samples were treated with RNase A for 30 min at 37°C before reversal of the crosslink with 200 mM NaCl for 5h at 65°C. DNA-protein mix was precipitated with 2.5x volume of 100% ethanol overnight at -20°C. The ethanol was removed after centrifugation for 15 min at 13,000 rpm 4°C. The pellets were resuspended in TE, 5x Proteinase K buffer (50 mM Tris–HCl pH 7.5, 25 mM EDTA, 1.25% SDS) and Proteinase K (20 mg/ml), and incubated for 2h at 45°C. DNA was purified with a QIAGEN PCR purification kit and kept at -20°C.

ChIP samples were analysed by real-time qPCR with a QuantiTect SYBR Green PCR kit (QIAGEN) and a Rotor-Gene RG-3000 (Corbett Research). Each reaction was performed with: 1 μl of template, 1 μl of primer pair mix (10 μM), 3 μl of water and 5 μl of SYBR Green Mix (2×). The sequences of primers used in this study can be found in Table 1. The thermo-cycling parameters were: 95°C for 15 min followed by 40 cycles of 94°C for 15 s, 57°C for 20 s and 72°C for 25 s. The Roto-Gene Q Series Software was used to calculate the threshold cycle

**Table 1. Sequences of primers used for PCR, ChIP-qPCR, and qRT-PCR.**

| Name | Forward primer | Reverse primer |
| --- | --- | --- |
| **PCR** | | |
| DNAJB1 E2/3 | GAACCAAAATCACTTTCCCCAAGGAAGG | AATGAGGTCCCCACGTTTCTCGGGTGT |
| BRD2 E4/5 | CAAAATTATAAAACAGCCTATGGACATG | TTTTCCAGCGTTTGTGCCATTAGGA |
| **ChIP-qPCR** | | |
| KPNB1 TSS | TTACTTCCTCCCTCCAAATGGG | ACAGCCTCCCTTCCTTCTTTC |
| KPNB1 TSS+3.2 | GCCCAGAGAACAAGAAATCG | GGAATGGACAAGCTGTGTTG |
| KPNB1 TSS+20.2 | TGCAAGAGCCAGTGGGAACACTT | CCTCTACTCAGCAATGATACTTC |
| KPNB1 pA-4.8 | CTGAGGAAACTGAAGAACCAAG | GAAGGCAGTGCTTGCCAGAAT |
| KPNB1 pA-2.9 | GAGGAGTGTGCACGGATGCTGAA | CCAAGATGGCCGATGTTATGG |
| KPNB1 pA-0.4 | TAGTTACCGTCTGCTTGGGAAGATG | CCTCTGACAGCAAGTCCAACATT |
| KPNB1 pA+1.4 | GACTCATCACACCAAGGTCAC | GATAGTGCTGGGAAGGAAATGG |
| KPNB1 pA+2.6 | GTACATCTCAGCTTTGGCATATG | GCCCAGAACATAGCAGGCATTGC |
| KPNB1 pA+4.1 | GTTTCACCGTGTTAGCCAGGATGG | CCACAGCCATGTTCATTTCTGC |
| KPNB1 TD (pA +11.3) | AGGAGCATGGCTTTTCTCTG | TCATGCTGGAACTGGTTGAG |
| KPNB1 +12.8 | TGGAAGTTCTGATGCTCCTG | TCAAGTTAATGCCCCTGCTC |
| JUN TSS | TTTAGGGGTTGACTGGTAGCAG | TGTCTGTCTGTCTGCCTGAC |
| JUN GB | TGACCGCGACTTTTCAAAGC | GCGCAGGGTTAATTAAGATGCC |
| JUN pA | ATCAAGTGGCATGTGCTGTG | TGCCACCAATTCCTGCTTTG |
| JUN TD | TCACCCCTTCATGTTTCTGC | GTGTCACTTACGCCAAGACTTG |
| H1-2 GB | ACTCTGGTGCAAACGAAAGG | TTAACCTTGGGCTTGGCTTC |
| H1-2 T1 | TCCCTCTGTTTGACATCCATGG | TGAGCAGCTTATCTCCAGAAGC |
| H1-2 T2 | TCACGGGTTCAAGCGATTCTC | AAAATTAGCCGGGCATGGTG |
| Neg | TGGTACAACCACAGCTCAGTG | AAGCTGGACATGGTTGTGTG |
| **qRT-PCR** | | |
| ATF3 | AACCACAGTCAGTGGAGAGATG | TTCTCACAGCTGCAAACACC |
| LDLR | TTTGACGGGACTTCAGGTTC | TCCCTTGTGACATCTTCACG |
| NR4A3 | AAGCCACCAGCTGTTAATGG | GCAATGCTGTTAGAGGAGCAG |
| PHLDB2 | GAAACGACTTCAGGCAAGTCTC | GCCATGTTTTAGGAAAGAGCAC |
| SLCO4A1 | TCCTCTTCTTTGCCATAGCC | ACAAGTTTCCAGGCCATCTG |
| ATP5F1B | ACCCATTGAAGAAGCTGTGG | CAATCAAGGCTCTTGTGCAG |
| TARS1 | CATGGAAAAGGAGGAACAGC | CTTTGCCAAACTCCTCCAAG |
| H1-2 | ACTCTGGTGCAAACGAAAGG | TTAACCTTGGGCTTGGCTTC |
| TUBB | ATATGTTCCTCGTGCCATCC | TTTGGCCCAGTTGTTACCTG |
| GAPDH | CAACGACCACTTTGTCAAGC | TTCCTCTTGTGCTCTTGCTG |
| KPNB1 (poly(A)+) | GAGGAGTGTGCACGGATGCTGAA | CCAAGATGGCCGATGTTATGG |
| KPNB1 (Random primers) | CAGTCTGTGATGGCATTTAAG | TCTGACTTTCAAACCATTTCACCTCC |
| JUN | TGACCGCGACTTTTCAAAGC | GCGCAGGGTTAATTAAGATGCC |
| 7SK | CTGATCTGGCTGGCTAGGCGGG | GAAGACCGGTCCTCCTCTATCGG |
| Pre-rRNA | CCTGCTGTTCTCTCGCGCGTCCGAG | AACGCCTGACACGCACGGCACGGAG |

(Ct) value. Signals are presented as a percentage of Input DNA after removal of the IgG background signal. Each ChIP sample was measured in triplicate by qPCR. The number of biological replicates is indicated on each figure.

## RNA preparation and PCR/qPCR

Total RNA was extracted from ~70% confluent HeLa cells grown in 6-wells plate using a RNeasy Micro Kit (Qiagen) with a DNase step according to the manufacturer's instructions.

Reverse-transcription (RT) was performed with 500 ng of RNA using random hexamers or oligo(dT) with the SuperScript III kit (Invitrogen) according to the manufacturer's instructions. PCR reactions were performed with the Phusion High-Fidelity DNA polymerase (NEB) according to the manufacturer's instructions. The sequence of primers used for PCR is given in Table 1. Quantifications were performed with Image Studio Lite version 5.2.5. The quantifications are shown as % of unspliced reads and calculated as follow: (unspliced RNA signal) / (spliced RNA signal + unspliced RNA signal) * 100.

cDNA was amplified by qPCR with a QuantiTect SYBR Green PCR kit (QIAGEN) and a Rotor-Gene RG-3000 (Corbett Research). The sequence of primers used for qPCR is given in Table 1. Values are normalized to the 7SK non-coding RNA (random primers), or the GAPDH mRNA (oligo d(T)), that have been used as control. Experiments were replicated at least three times to ensure reproducibility, and each RNA sample was measured in triplicate by qPCR.

## 5'EU

HeLa cells were seeded at 40–50% confluency on coverslips in a 6-well plate, then treated the following day with Madrasin or Isoginkgetin for the time indicated on the figure before being incubated with 1 mM 5-ethylnyl uridine (5-EU) for 1 h to label newly synthesized RNA. Cells were then fixed by transferring coverslips to 3.7% formaldehyde diluted in PBS for 15 min, washed with PBS, and then permeabilised in 0.5% Triton X-100 diluted in PBS for 15 min at room temperature. Global nascent RNA transcription was detected using Click-iT® RNA Alexa Fluor 488 Imaging kit (ThermoFisher, C10329) according to the manufacturer's instructions. Following Click-iT reaction, cellular DNA was immunostained with Hoechst 33342 diluted 1:1000 in PBS for 15 min at room temperature, protected from light. Coverslips were then washed twice with PBS, then mounted onto slides using Fluoromount-G (SouthernBiotech, 0100–01) containing DAPI. Slides were visualised with an inverted multi-channel fluorescence microscope (Evos M7000; ThermoFisher). Fluorescent intensity was analysed and quantified with ImageJ (version 1.53q) by first using the DAPI channel (λEx/λEm (with DNA) = 350/461 nm) image to define the nucleus, then GFP channel (λEx/λEm = 495/519 nm) to measure the fluorescent intensity within the nucleus. Data were plotted by box-and-whisker plot with GraphPad Prism 9.5 software with the following settings: boxes: 25–75 percentile range; whiskers: min-max values; horizontal bars: median.

## mNET-seq

mNET-seq was carried out as previously described [32]. In brief, the chromatin fraction was isolated from $4 \times 10^7$ HeLa cells treated with DMSO or 90 μM Madrasin for 30 min. Chromatin was digested in 100 μl of MNase (40 units/μl) reaction buffer for 2 min at 37°C in a thermomixer at 1,400 rpm. MNase was inactivated with the addition of 10 μl EGTA (25 mM) and soluble digested chromatin was collected after centrifugation at 13 000 rpm for 5 min at 4°C. The supernatant was diluted with 400 μl of NET-2 buffer and antibody-conjugated beads were added. Antibodies used: Pol II (MABI0601, MBL International) and Ser5P (MABI0603, MBL International). Immunoprecipitation was performed at 4°C on a rotating wheel at 16 rpm for 1h. The beads were washed six times with 1 ml of ice-cold NET-2 buffer and once with 100 μl of 1x PNKT (1x PNK buffer and 0.05% Triton X-100) buffer. Washed beads were incubated in 200 μl PNK reaction mix for 6 min in a Thermomixer at 1,400 rpm at 37°C. The beads were then washed once with 1 ml of NET-2 buffer and RNA was extracted with Trizol reagent. RNA was suspended in urea Dye and resolved on a 6% TBU gel (Invitrogen) at 200 V for 5 min. In order to size select 35–100 nt RNAs, a gel fragment was cut between BPB and XC dye markers.

For each sample, several small holes were made with a 25G needle at the bottom of a 0.5 ml tube, which is then placed in a 1.5 ml tube. Gel fragments were placed in the layered tube and broken down by centrifugation at 12,000 rpm for 1 min. The small RNAs were eluted from the gel using RNA elution buffer (1 M NaOAc and 1 mM EDTA) for 1h on a rotating wheel at 16 rpm at room temperature. Eluted RNAs were purified with SpinX column (Coster) and two glass filters (Millipore) and the resulting flow-through RNA was precipitated overnight with ethanol. RNA libraries were prepared according to manual of Truseq small RNA library prep kit (Illumina). Deep sequencing (Hiseq4000, Illumina) was conducted by the high throughput genomics team of the Wellcome Trust Centre for Human Genetics (WTCHG), Oxford.

## Bioinformatics analyses

**Transcription units' annotation.** Gencode V35 annotation, based on the GRCh38 version of the human genome, was used to obtain the list of protein-coding genes as in [8]. To obtain the list of expressed mRNA isoforms, Salmon version 0.14.1 [33] was used on two HeLa nucleoplasm RNA-seq (GSE110028 [34]) and the most expressed mRNA isoform of each gene was kept for further analysis. HeLa chromatin RNA-seq signal (GSE110028 [34]) was then quantified with BEDtools version 2.29.2 [35] multicov to keep only the most expressed mRNA isoform with more than 10 reads in two biological replicates.

For the intron-containing, intronless, and histone genes, the mRNAs isoforms were separated in 193 intronless genes, 40 histone genes, and 2,236 intron-containing genes with a maximal gene size corresponding to the size of the longest intronless gene. The list of used exons was extracted from Gencode V35 by using the annotation of each most expressed mRNA isoform. The first and final exons of each mRNA isoform were then removed to keep only internal exons.

**mNET-seq analysis.** Adapters were trimmed with Cutadapt version 1.18 [36] in paired-end mode with the following options:—minimum-length 10 -q 15,10 -j 16 –A `GATCGTCGGA CTGTAGAACTCTGAAC` –a `AGATCGGAAGAGCACACGTCTGAACTCCAGTCAC`. Trimmed reads were mapped to the human GRCh38.p13 reference sequence with STAR version 2.7.3a [37] and the parameters:—runThreadN 16—readFilesCommand gunzip -c -k—limitBAMsortRAM 20000000000—outSAMtype BAM SortedByCoordinate. SAMtools version 1.9 [38] was used to retain the properly paired and mapped reads (-f 3). A custom python script [32] was used to obtain the 3′ nucleotide of the second read and the strandedness of the first read. Strand-specific bam files were generated with SAMtools. FPKM-normalized bigwig files were created with deepTools2 version 3.4.2 [39] bamCoverage tool with the parameters -bs 1 -p max—normalizeUsing RPKM.

## RNA-seq analysis

RNA-seq in HeLa cells treated for 12h with DMSO or 35 μM Isoginkgetin were taken from GSE72055 [40]. Total RNA-seq in HeLa cells for 6h or 18h treatment with DMSO or 30 μM Isoginkgetin were taken from GSE86857 [30]. RNA-seq were analysed as in [41]. Adapters were trimmed with Cutadapt in paired-end mode with the following options:—minimum-length 10 -q 15,10 -j 16 -a `AGATCGGAAGAGCACACGTCTGAACTCCAGTCA` -A `AGATCGGAA GAGCGTCGTGTAGGGAAAGAGTGT`. The remaining rRNA reads were removed by mapping the trimmed reads to the rRNA genes defined in the Human ribosomal DNA complete repeating unit (GenBank: U13369.1) with STAR and the parameters—runThreadN 16—readFilesCommand gunzip -c–k—outReadsUnmapped Fastx—limitBAMsortRAM 20000000000—outSAMtype BAM SortedByCoordinate. The unmapped reads were mapped to the human GRCh38.p13 reference sequence with STAR and the and the ENCODE parameters:—runThreadN 16—limitBAMsortRAM 20000000000—outSAMtype BAM SortedByCoordinate—

quantMode GeneCounts—outFilterMultimapNmax 20—outFilterType BySJout—alignSJover-hangMin 8—alignSJDBoverhangMin 1—outFilterMismatchNmax 999—alignIntronMin 20—alignIntronMax 1000000—alignMatesGapMax 1000000. SAMtools was used to retain the properly paired and mapped reads (-f 3) and to produce strand-specific bam files. HTSeq version 1.99.2 [42] count tool and the Gencode v35 annotation has been used to obtain the number of counts per gene. DESeq2 version 1.30.1 [43] was then used to perform the differential expression analysis. Any gene with a two-fold fold change and an adjusted p-value < 0.05 was considered as significant. FPKM-normalized bigwig files were created with deepTools bam-Coverage tool with the parameters -bs 10 -p max—normalizeUsing RPKM.

**Splicing efficiency.** RNA-seq splicing efficiency was calculated as previously [8] by first parsing each bam file to obtain the list of spliced and unspliced reads with the awk command (awk '/^@/|| $6 ~ /N/' for spliced reads and awk '/^@/|| $6! ~ /N/' for unspliced reads). The splicing efficiency was then calculated as the number of spliced reads over total reads for each expressed protein-coding gene with BEDtools multicov–s–split.

**Metagene profiles.** Metagene profiles of genes scaled to the same length were generated with deepTools2 computeMatrix tool with a bin size of 10 bp and the plotting data obtained with plotProfile–outFileNameData tool. Graphs representing the RNA-seq or the mNET-seq signal were then created with GraphPad Prism 9.5. Metagene profiles are shown as the average of two biological replicates.

**P values and significance tests.** P-values were computed with unpaired two-tailed Student's t test, Wilcoxon rank sum test, or Wilcoxon signed-rank test, as indicated in the legends. Statistical tests were performed in GraphPad Prism 9.5.

## Results

### Isoginkgetin and Madrasin are poor splicing inhibitors

We first determined whether a short treatment with Madrasin or Isoginkgetin (Fig 1A) inhibits pre-mRNA splicing in HeLa cells as observed for PlaB and HB [7, 21]. As the effect of Madrasin is less well characterised than Isoginkgetin, we first determined which concentration of Madrasin would be optimal for short-term inhibition. We treated HeLa cells with 30 μM, 60 μM, or 90 μM Madrasin for 30 min followed by RT-PCR analysis (S1A Fig). We observed a concentration-dependent inhibition of splicing of the intron located between exons 4 and 5 of *BRD2* and selected 90 μM as a working concentration. To compare the efficiency of the different splicing inhibitors, we treated HeLa cells for 1h with DMSO, 1 μM PlaB, 1 μM HB, 30 μM Isoginkgetin, or 90 μM Madrasin and performed RT-PCR for exons 2 and 3 of *DNAJB1* and exons 4 and 5 of *BRD2* (Fig 1B). While SF3B1 inhibition with PlaB and HB shows, as expected, a clear increase in intron retention for both genes, Isoginkgetin and Madrasin have little effect on pre-mRNA splicing of these two genes. As SF3B1 inhibition also affects transcription [5, 7, 19, 21], we investigated nascent transcription activity via 5'EU incorporation in HeLa cells treated for 1h with DMSO, 100 μM DRB, a CDK9 inhibitor, 1 μM PlaB, 1 μM HB, 30 μM Isoginkgetin, or 90 μM Madrasin (Fig 1C and 1D). As we previously observed, CDK9 and SF3B1 inhibition results in a reduction in 5'EU incorporation [21] whereas Madrasin, but not Isoginkgetin, decreases the 5'EU signal. To determine whether the decrease in 5'EU signal is due to a degradation of pol II, we fractionated into chromatin, nucleoplasm, and cytoplasm fractions HeLa cells treated for 1h with DMSO, 1 μM PlaB, 1 μM HB, 30 μM Isoginkgetin, or 90 μM Madrasin and probed for total pol II, SF3B1, and α-tubulin and histone H3 as loading controls (Fig 1E). Madrasin and SF3B1 inhibition with PlaB and HB decrease SF3B1 chromatin association but do not affect the level of Rpb1, whereas Isoginkgetin does not affect SF3B1 or Rpb1 levels.

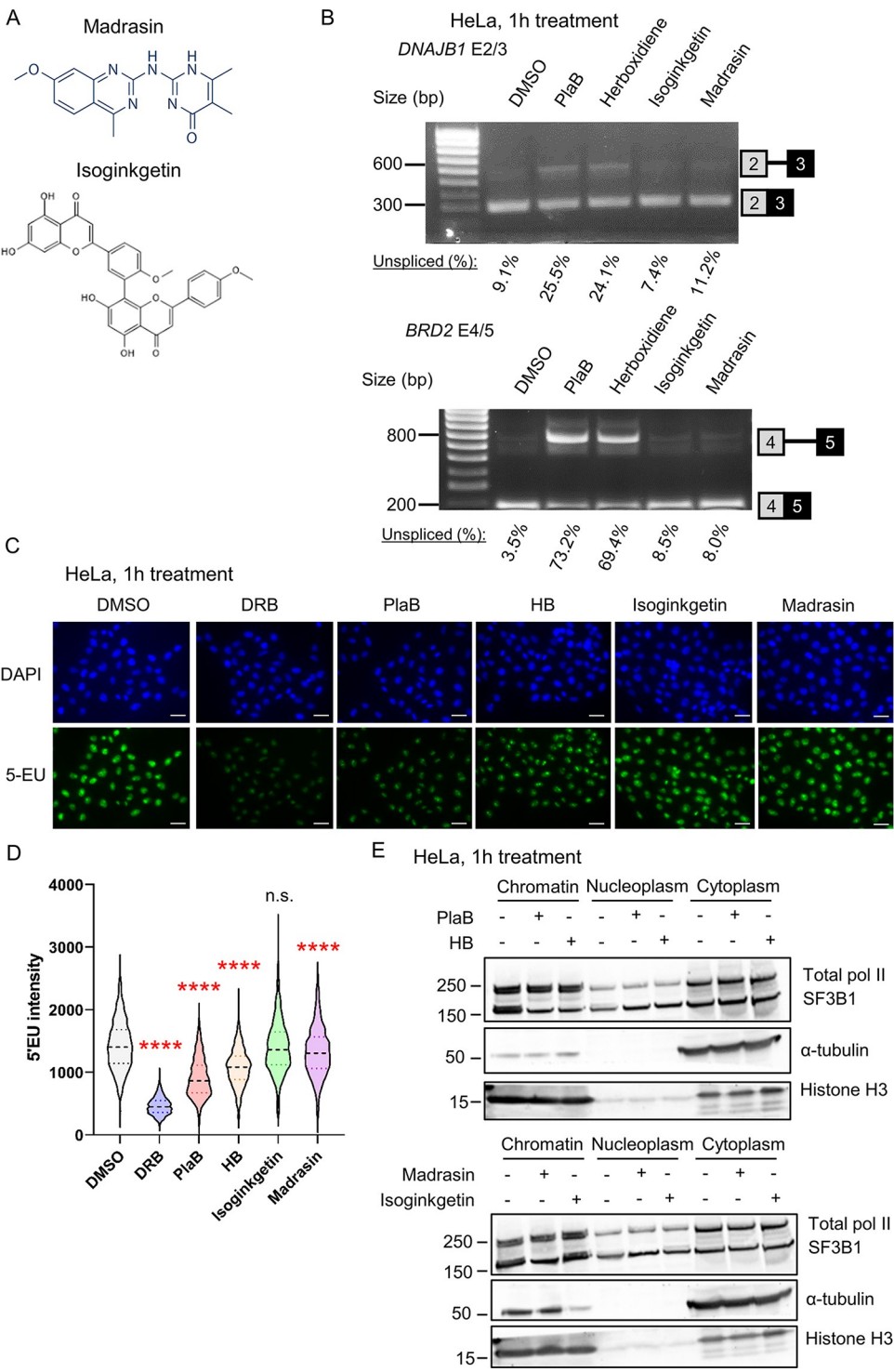

**Fig 1. Isoginkgetin and Madrasin are poor splicing inhibitors. (A)** Chemical structure of Madrasin and Isoginkgetin. **(B)** RT-PCR with primers amplifying the intron located between exons 2 and 3 of *DNAJB1* or exons 4 and 5 of *BRD2*. HeLa cells were treated with DMSO, 1 μM PlaB, 1 μM HB, 30 μM Isoginkgetin, or 90 μM Madrasin for 1h. The location of the spliced and unspliced RNA is shown on the left of the panel. The percentages of unspliced RNA compared to total (spliced + unspliced) are shown below. **(C)** Representative images of immunofluorescence analysis or 5'EU incorporation in HeLa cells treated with DMSO, 100 μM DRB, a CDK9 inhibitor, 1 μM PlaB, 1 μM HB, 30 μM Isoginkgetin, or 90 μM Madrasin for 1h. EU (green), DAPI (blue), scale bars: 50 μm. **(D)** Quantification of 5'EU intensity per nucleus for DMSO, DRB, PlaB, HB, Isoginkgetin, and Madrasin. Boxplot settings are: min to max values

with the box showing 25–75 percentile range. > 10,000 nuclei were quantified per condition. Statistical test: Kruskal-Wallis test. P-value: n.s. not significant, **** < 0.0001. (**E**) Western blots of total pol II, SF3B1, α-tubulin (loading control), and histone H3 (loading control) from chromatin, nucleoplasm, and cytoplasm fractions of HeLa cells treated with DMSO, 1 μM PlaB, 1 μM HB, 30 μM Isoginkgetin, or 90 μM Madrasin for 1h.

These results indicate that both Madrasin and Isoginkgetin are poor splicing inhibitors but that Madrasin affects transcription without degrading pol II.

## Madrasin decreases transcription of protein-coding genes

As Madrasin reduces 5'EU signal, we first investigated by qRT-PCR whether rDNA transcription is decreased after 30 min or 60 min treatment with 90 μM Madrasin (S1B Fig). As the pre-rRNA level is not affected by Madrasin, we investigated pol II nascent transcription by performing total pol II and Ser5-P mNET-seq in HeLa cells following 30 min treatment with DMSO (Control) or 90 μM Madrasin (Fig 2A and 2B and S1C Fig). Madrasin treatment causes a decrease in pol II pausing and in pol II levels across the gene body on protein-coding genes while the pol II level increases downstream of poly(A) sites, indicating a transcription termination defect. We confirmed the decrease in pol II transcription following Madrasin treatment in mNET-seq by qRT-PCR on eight protein-coding genes (Fig 2C) and by pol II ChIP-qPCR on our model protein-coding gene *KPNB1* (Fig 2D and 2E). To understand whether the effect of Madrasin on transcription could be mediated by a loss of transcription elongation and termination factors, we performed ChIP-qPCR on the *KPNB1* gene of the elongation factors SPT5 and CDC73, a subunit of the PAF1 complex, and of CPSF73, the pre-mRNA cleavage enzyme, in HeLa cells treated with DMSO or 90 μM Madrasin for 30 min (Fig 2E and 2F). When the results are not ratioed to pol II (Fig 2E), the effect of Madrasin on SPT5 and CDC73 mimics the effect on pol II, with a reduction across the whole gene body and an increase in the delayed transcription termination region and the CPSF73 signal is not affected by Madrasin treatment. When ratioed to pol II, there is a relative increase in SPT5, CDC73, and CPSF73 across the gene body of *KPNB1* but a reduced level of the three proteins on the delayed transcription termination region following Madrasin treatment (Fig 2F).

These results indicate that while Madrasin has a limited effect on pre-mRNA splicing, it downregulates pol II transcription of protein-coding genes and causes a transcription termination defect.

## Transcription of intronless and histone genes is also affected by Madrasin

As Madrasin promotes a general transcriptional defect on protein-coding genes, which also include intronless and histone genes, we have analysed the effect of this drug on intronless genes with poly(A) sites and on intronless replication-activated histone genes. As intron-containing genes are generally longer than intronless genes, we selected three groups of expressed genes in HeLa cells: histone genes (n = 40), intronless genes (n = 193), and intron-containing protein-coding genes with a gene size inferior or equal to the longest intronless genes (n = 2,236) (S2A Fig). The RNA level from the intron-containing and intronless genes is similar in chromatin RNA-seq, while histone genes produce more RNA (S2B Fig). As expected, transcription of intron-containing genes is similarly affected by Madrasin as all protein-coding genes (S2C Fig). Madrasin also decreases nascent transcription of intronless genes and causes a higher pol II signal downstream of the poly(A) site (Fig 3A and S2D Fig). Interestingly, Madrasin activates the expression of some intronless genes, such as *JUN* (Fig 3B). We confirmed the increase in pol II, SPT5, and CDC73, but not CPSF7, on *JUN* after Madrasin treatment by ChIP-qPCR (Fig 3C and S2E Fig). Histone genes are also affected by Madrasin,

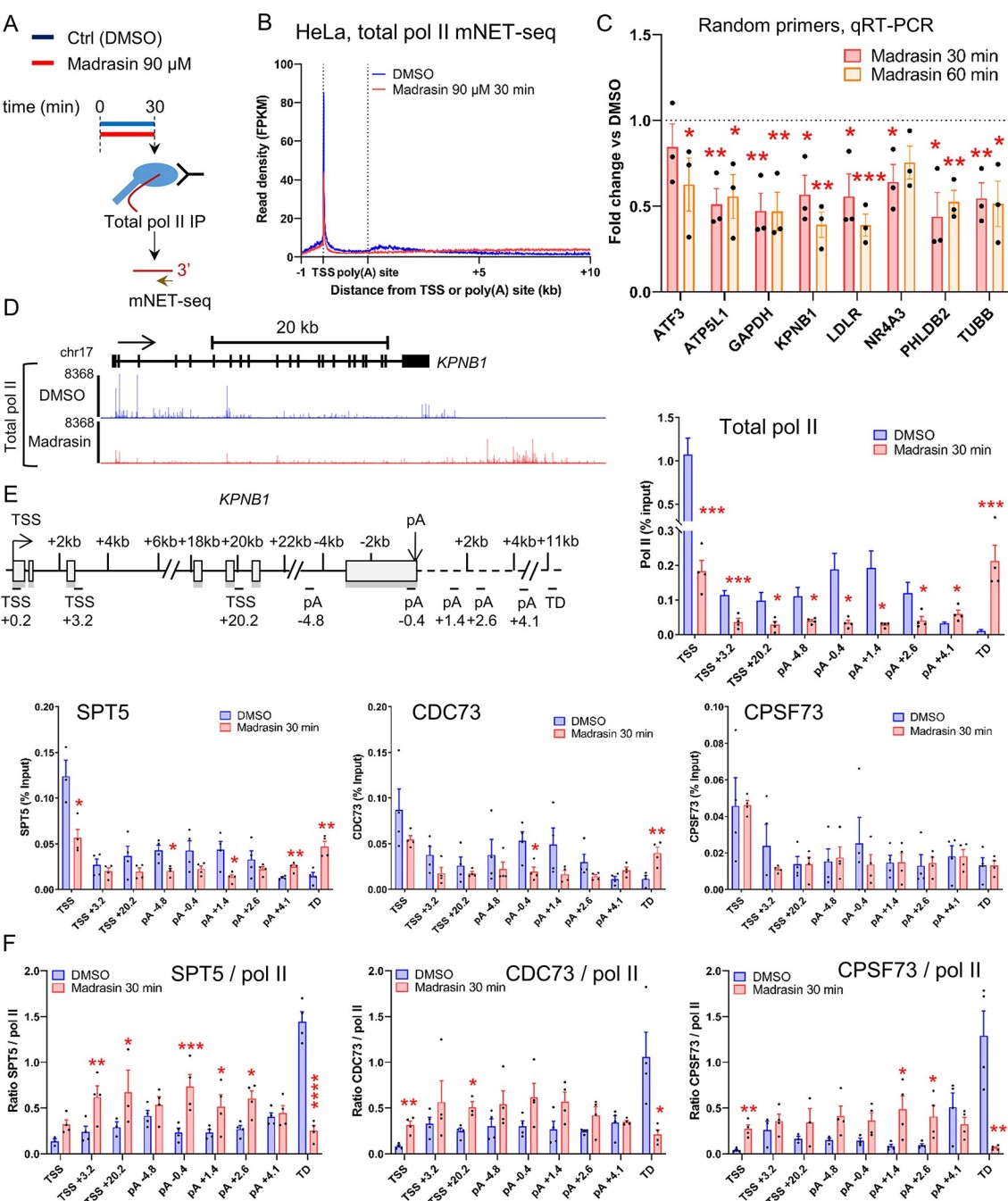

**Fig 2. Madrasin decreases transcription of protein-coding genes. (A)** Schematic of the mNET-seq technique. **(B)** Metagene profile of total pol II mNET-seq treated with DMSO (blue) or 90 μM Madrasin (red) for 30 min on scaled expressed protein-coding genes. **(C)** qRT-PCR with primers amplifying different protein-coding genes. HeLa cells were treated with DMSO or 90 μM Madrasin for 30 min (red) or 60 min (orange). cDNA was generated with random hexamers. Values are normalised to the 7SK snRNA and shown as relative to DMSO, mean ± SEM, n = 3 biological replicates. Statistical test: two-tailed unpaired t-test. P-value: * < 0.05, ** < 0.01, *** < 0.001. **(D)** Screenshot of the genome browser total pol II mNET-seq DMSO (blue) and Madrasin (red) tracks on the protein-coding gene *KPNB1*. The arrow indicates the sense of transcription. **(E)** Total pol II, SPT5, CDC73, and CPSF73 ChIP-qPCR across the protein-coding gene *KPNB1* in HeLa cells treated with DMSO (blue) or 90 μM Madrasin for 30 min (red). Mean ± SEM, n = 3 biological replicates. Statistical test: two-tailed unpaired t-test. P-value: * < 0.05, ** < 0.01, *** < 0.001. **(F)** Ratios of SPT5 / total pol II, CDC73 / total pol II, or CPSF73 / total pol II from ChIP-qPCR on the intron-containing gene *KPNB1* in HeLa cells treated with DMSO (blue) or 90 μM Madrasin for 30 min (red). Mean ± SEM, n = 3 biological replicates. Statistical test: two-tailed unpaired t-test. P-value: * < 0.05, ** < 0.01, *** < 0.001, **** < 0.0001.

mostly promoting a higher pol II signal downstream of the histone genes without affecting the level of nascent transcripts across the coding region (Fig 3D and 3E and S2F Fig). While pol II ChIP-qPCR on the *H1-2* gene confirmed the mNET-seq profile, Madrasin treatment reduces SPT5 and CDC73, but not CPSF73, on this histone gene (Fig 2F and S2G Fig).

These results indicate that Madrasin globally affects pol II transcription on intron-containing and intronless genes, and in the case of histone genes, mostly on transcription downstream of the normal transcription termination region.

## Isoginkgetin treatment affects pol I and pol II transcription

While a 1h treatment with 30 μM Isoginkgetin did not affect pre-mRNA splicing and 5'EU incorporation (Fig 1B–1D), we investigated whether a longer treatment time would result in a change in splicing and transcription as Isoginkgetin was already shown to affect transcription after 6h of treatment [30]. We treated HeLa cells with 30 μM Isoginkgetin from 1h to 6h and found virtually no effect on pre-mRNA splicing as analysed by RT-PCR on *DNAJB1* and *BRD2* genes (Fig 4A). Re-analysis of previously-published total RNA-seq datasets [30, 40] in HeLa cells following 6h, 12h, or 18h treatment with 30 μM Isoginkgetin confirms that pre-mRNA splicing is not affected by Isoginkgetin at 6h but is significantly decreased at 12h and 18h (Fig 4B). Isoginkgetin is therefore unlikely to act on pre-mRNA splicing as a primary effect, in agreement with [30]. To test whether Isoginkgetin affects transcription prior to 6h treatment, we first monitored 5'EU incorporation in HeLa cells following a time-course of 1h to 6h treatment with 30 μM Isoginkgetin (Fig 4C and 4D). It takes approximately 5h of Isoginkgetin treatment to see a marked decrease in 5'EU incorporation while shorter treatment time points show rather an increase in 5'EU incorporation. To determine whether pre-rRNA transcription could be responsible for the change in 5'EU incorporation, we investigated whether Isoginkgetin affects pol I transcription by qRT-PCR (Fig 4E). Transcription of rDNA starts to decrease after 2h treatment with Isoginkgetin but becomes significantly reduced only at 4h and 5h.

We next investigated by oligo d(T) qRT-PCR the expression of 11 protein-coding genes in HeLa cells treated with DMSO or 30 μM Isoginkgetin from 1h to 6h (Fig 5A). We observed a detectable effect on the production of poly(A)+ mRNA from 3h onwards (*NR4A3*, *LDLR*, and *KPNB1*), with generally a downregulation of protein-coding gene expression, with the exception of the intronless gene *JUN*, following Isoginkgetin treatment. A general downregulation of expression of protein-coding genes by this drug is also apparent after re-analysis of previously published RNA-seq datasets (6h, 12h, and 18h Isoginkgetin treatment), which shows that protein-coding genes are generally downregulated while non-coding genes tend to be upregulated (S3A and S3B Fig). *ATF3* expression was previously shown to be upregulated in HeLa cells after a 24h treatment with Isoginkgetin [44], while we observed a decrease in ATF3 expression after 6h treatment (Fig 5A). We performed qRT-PCR following a 24h treatment with Isoginkgetin and confirmed that both *JUN* and *ATF3* are upregulated while four other protein-coding genes are downregulated (S3C Fig). As changes in the level of poly(A)+ mRNA take more time to appear than changes in transcription, we performed pol II ChIP-qPCR following Isoginkgetin treatment (Fig 5B–5D and S3D Fig). We did not observe any change in total pol II across *KPNB1* after treating HeLa cells with Isoginkgetin for 30 min (S3D Fig). However, an Isoginkgetin time-course from 1h to 6h shows a decrease in pol II signal from 2h onwards across *KPNB1*, *JUN*, and *H1-2* (Fig 5B–5D), which would explain the downregulation in poly(A)+ mRNA production apparent from 3h treatment onwards.

To determine whether Isoginkgetin could affect pol II CTD phosphorylation before transcription, we purified the chromatin fraction of HeLa cells treated with DMSO or 30 μM

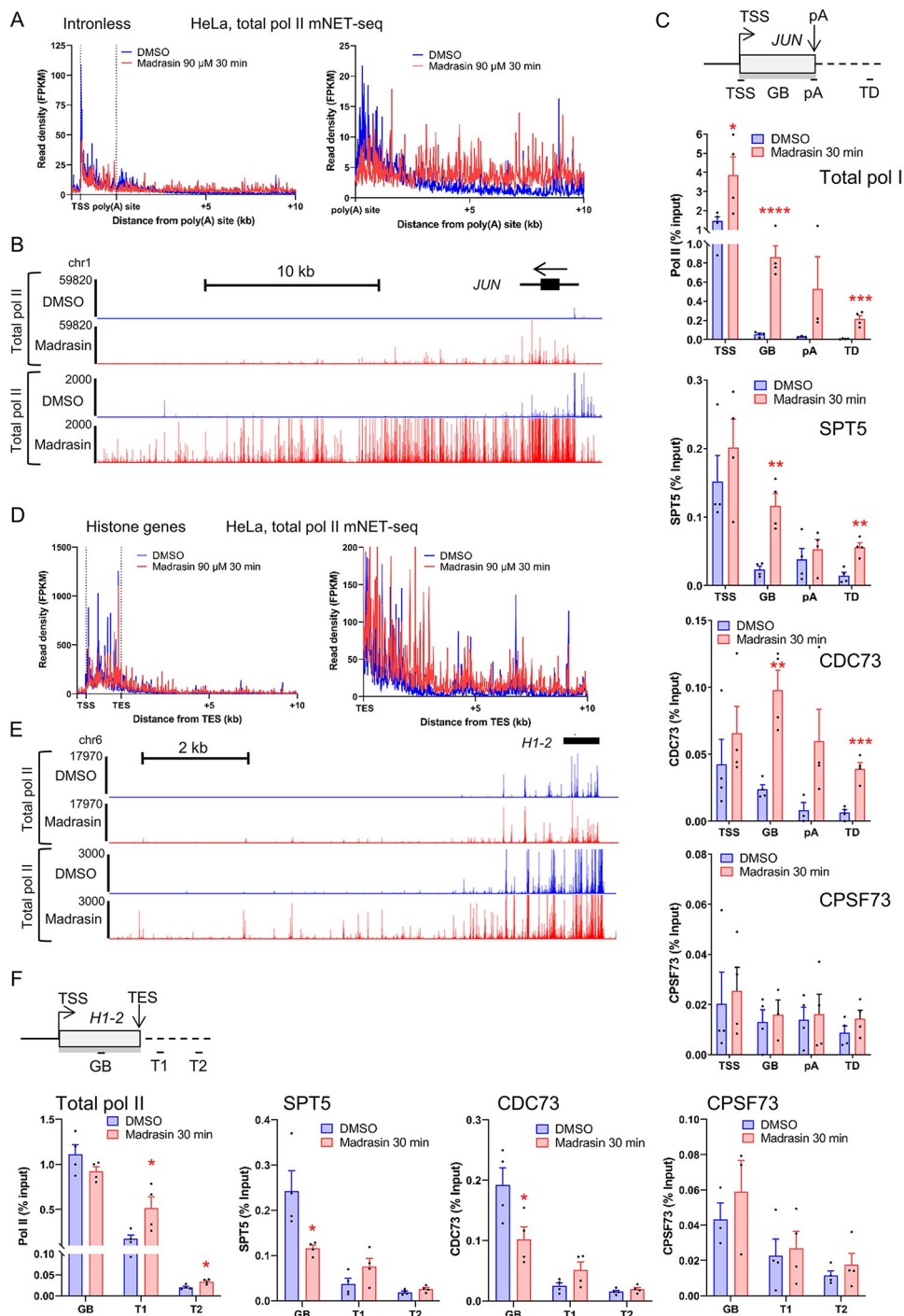

**Fig 3. Transcription of intronless and histone genes is also affected by Madrasin.** (A) Metagene profiles of total pol II mNET-seq performed in HeLa cells treated with DMSO (blue) or 90 μM Madrasin for 30 min (red) on scaled expressed intronless genes. (B) Screenshot of the genome browser total pol II mNET-seq DMSO (blue) and Madrasin (red) tracks of the intronless gene *JUN*. The arrow indicates the sense of transcription. (C) Total pol II, SPT5, CDC73, and CPSF73 ChIP-qPCR on the intronless gene *JUN* in HeLa cells treated with DMSO (blue) or 90 μM Madrasin for 30 min (red). Mean ± SEM, n = 3 biological replicates. Statistical test: two-tailed unpaired t-test. P-value: * < 0.05, ** < 0.01, *** < 0.001, **** < 0.0001. (D) Metagene profiles of total pol II mNET-seq performed in HeLa cells treated with DMSO (blue) or 90 μM Madrasin for 30 min (red) on scaled expressed histone genes. (E) Screenshot of the genome browser total pol II mNET-seq DMSO (blue) and Madrasin (red) tracks of the histone gene *H1-2*. The arrow indicates the sense of transcription. (F) Total pol II, SPT5, CDC73, and CPSF73 ChIP-qPCR on the histone gene *H1-2*

in HeLa cells treated with DMSO (blue) or 90 μM Madrasin for 30 min (red). Mean ± SEM, n = 3 biological replicates. Statistical test: two-tailed unpaired t-test. P-value: * < 0.05.

Isoginkgetin for 1h, 2h, or 4h and performed western blots for total pol II, Ser2-P, and Ser5-P (Fig 5E). We generally observed a loss of total pol II and of Ser2 and Ser5 phosphorylation on chromatin over time, with the exception of an increase in Ser5-P at the 1h timepoint. To confirm these observations, we performed pol II, Ser2-P, and Ser5-P ChIP-qPCR across the *KPNB1* gene following Isoginkgetin treatment for 1h, 1h30, and 2h (Fig 5F). The changes in total pol II profile looks similar to what we previously observed after Isoginkgetin treatment (Fig 5B) and the changes in Ser2-P profile match the changes in total pol II, indicating that this mark is largely unaffected. However, the Ser5-P level increases at the promoter region after 1h treatment but returns to control levels after 1h30, in agreement with the chromatin western blot.

Isoginkgetin is a bioflavonoid, a group of small molecules that have been shown to regulate the SUMO post-translational modification [45]. The SUMO modification is present on mRNA splicing and CPA factors and regulates their activities [45–47]. To determine whether 1h, 2h, or 4h treatment of 30 μM Isoginkgetin modifies SUMO1 or SUMO2/3 levels, we performed western blots on HeLa whole cell extract (S3E Fig). We only observed a slight increase in total SUMO2/3 level after 4h treatment.

These results indicate that Isoginkgetin affects pol I and pol II transcription and pol II CTD phosphorylation rather than pre-mRNA splicing. The effect of Isoginkgetin on transcription is also not likely mediated via a global change in SUMOylation.

## Discussion

We set out to test whether inhibition of pre-mRNA splicing affects transcription elongation and transcription termination using two small molecules, Madrasin and Isoginkgetin, that were found to inhibit pre-mRNA splicing following a 24h treatment [28, 29]. To limit confounding effects associated with long treatment times, we have carried out short time courses of treatment. However, we rather found that both Madrasin and Isoginkgetin are poor splicing inhibitors and should therefore not considered as primarily inhibitors of pre-mRNA splicing.

While Madrasin has a limited effect on pre-mRNA splicing following a 30 min treatment, this small molecule promotes a general transcriptional downregulation and a delay in transcription termination, which is observed on intron-containing, intronless, and histone genes. The transcriptional dysregulation resulting from Madrasin could potentially explain the longer-term indirect effect on splicing via changes in proteins interacting with pol II, as we observed for the elongation factors SPT5 and CDC73. Another non-mutually exclusive possibility is a higher pol II elongation rate following Madrasin treatment, which could also explain the delayed transcription termination, as pol II elongation rate is known to regulate RNA processing [2–4, 6, 16, 48]. However, the cellular target(s) of Madrasin is unknown.

In line with a previous report [30], our results clearly indicate that Isoginkgetin is not a potent pre-mRNA splicing inhibitor. Splicing defects take more than 6h of Isoginkgetin treatment to appear, strongly suggesting that this is a secondary effect. However, Isoginkgetin affects pol II transcription and pol II CTD Ser2 and Ser5 phosphorylation in ~1h and pol I transcription in ~2h. Additionally, Isoginkgetin can also affect post-transcriptional regulation, as exemplified by the *JUN* gene. While we observed a clear increase in *JUN* poly(A)+ mRNA, transcription of *JUN* by pol II was decreased by Isoginkgetin, indicating a stabilisation of the *JUN* mRNA. The re-analysis of Isoginkgetin RNA-seq datasets also shows a clear enrichment of non-coding (nc)RNAs in upregulated genes, with ncRNAs representing approximately 2/3

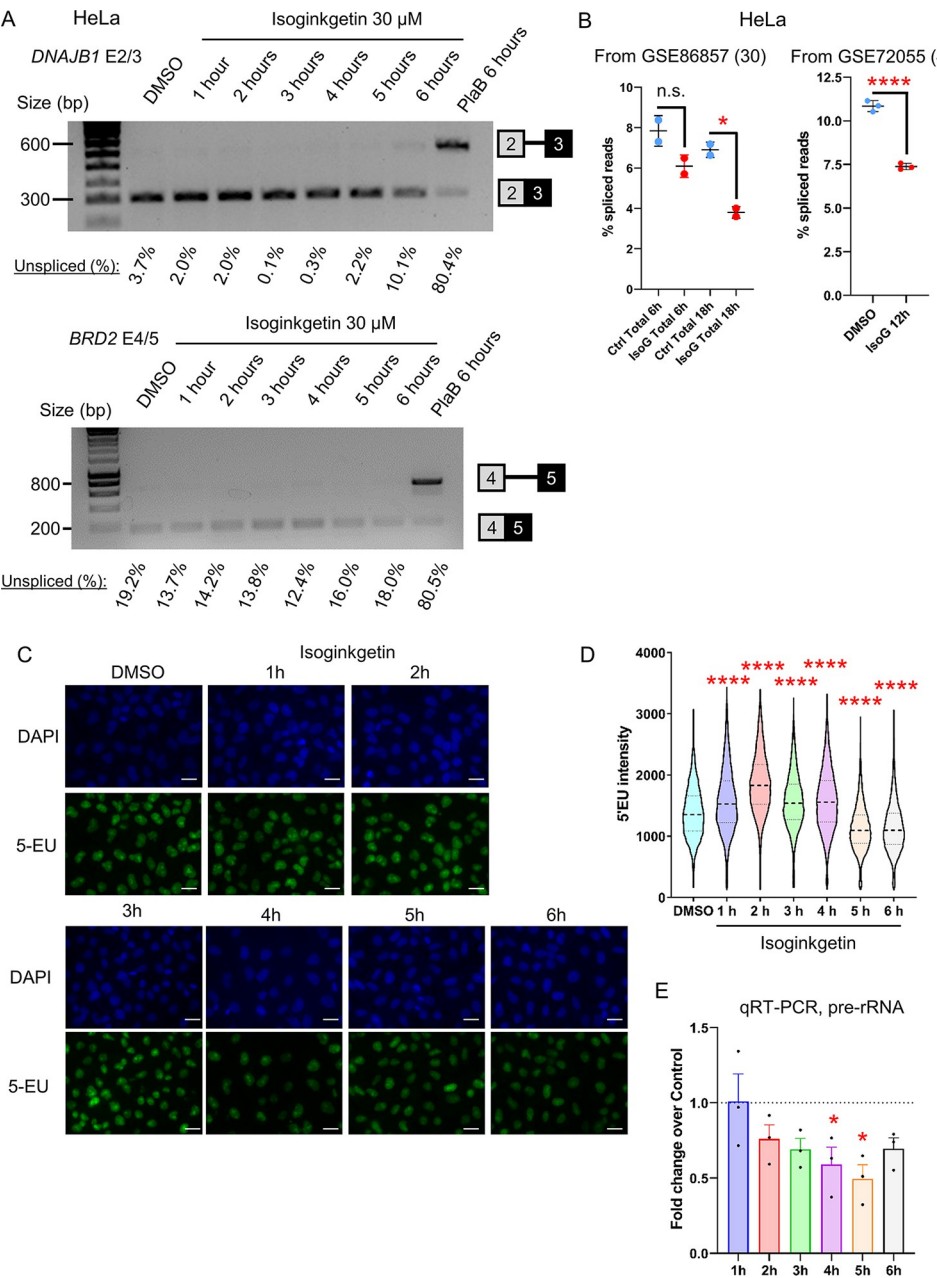

**Fig 4. Isoginkgetin is a poor splicing inhibitor.** (**A**) RT-PCR with primers amplifying the intron located between exons 2 and 3 of *DNAJB1* or the intron located between exons 4 and 5 of *BRD2*. HeLa cells were treated with DMSO, 30 μM Isoginkgetin for 1h to 6h, or 1 μM PlaB for 6h. The locations of the spliced and unspliced RNA are shown on the right of the panel. The percentages of unspliced RNA compared to total (spliced + unspliced) are shown below. (**B**) Percentage of spliced reads in total RNA-seq treated with DMSO (blue) or Isoginkgetin (red) for the time indicated on the figure. Each point represents a biological replicate. Statistical test: Wilcoxon rank sum test. P-value: n.s. not significant, $^* < 0.05$, $^{****} < 0.0001$. (**C**) Representative images of immunofluorescence analysis or 5'EU incorporation in HeLa cells treated with DMSO or 30 μM Isoginkgetin for 1h to 6h. EU (green), DAPI (blue), scale bars: 50 μm. (**D**) Quantification of 5'EU intensity per nucleus for DMSO and Isoginkgetin. Boxplot settings are: min to max values with the box showing 25–75 percentile range. 8,466 nuclei were quantified per condition. Statistical test: Kruskal-Wallis test. P-value: $^{****} < 0.0001$. (**E**) qRT-PCR with primers amplifying a region from the pol I transcribed pre-rRNA. HeLa cells were treated with DMSO or 30 μM Isoginkgetin for 1h to 6h. cDNA was generated with random hexamers. Values are normalised to the *7SK* snRNA and shown as relative to DMSO, mean ± SEM, n = 3 biological replicates. Statistical test: two-tailed unpaired t-test. P-value: n$^* < 0.05$.

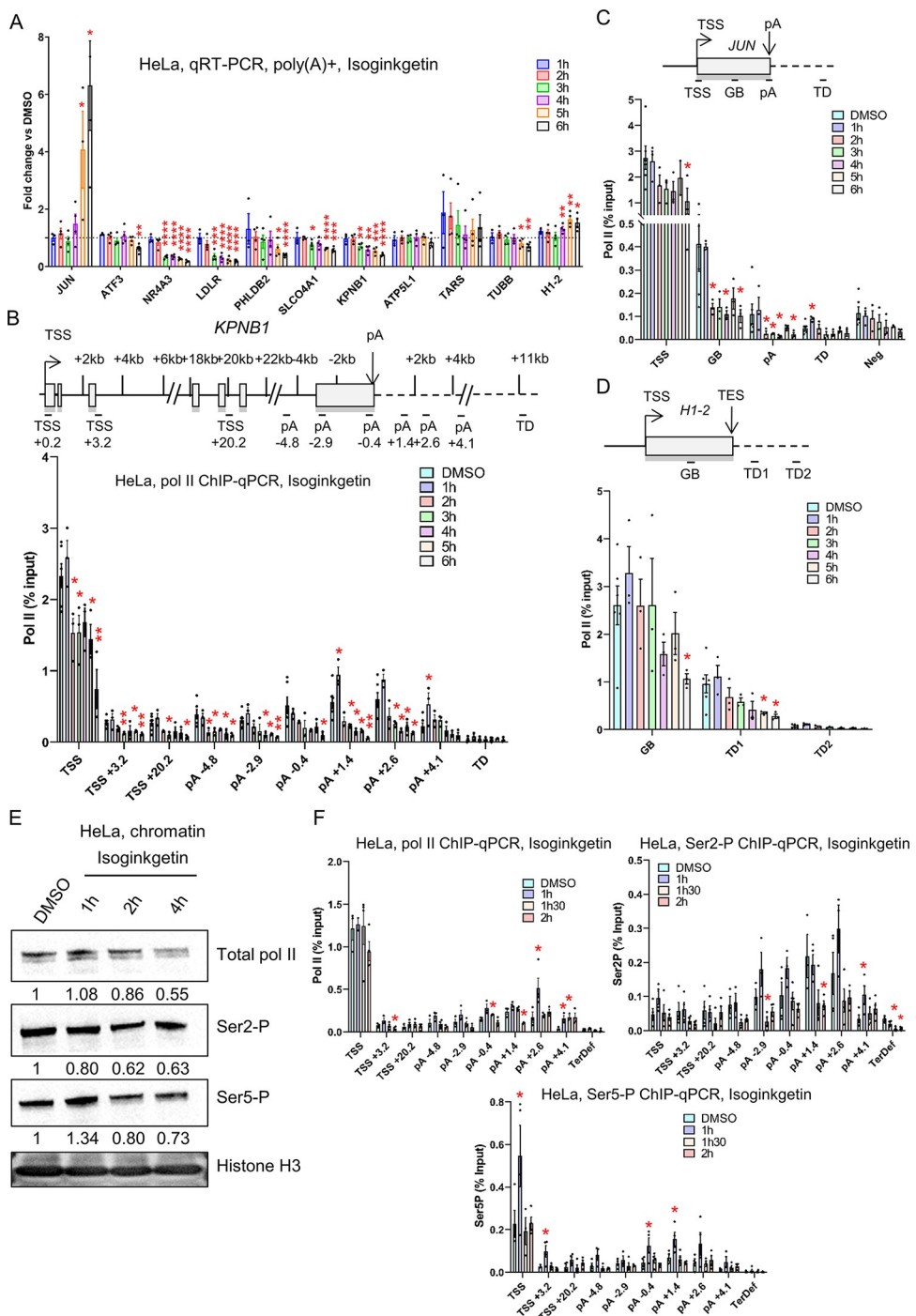

**Fig 5. Isoginkgetin treatment affects pol I and pol II transcription.** (**A**) qRT-PCR with primers amplifying different protein-coding genes. HeLa cells were treated with DMSO or 30 μM Isoginkgetin for 1h to 6h. cDNA was generated with oligo(dT). Values are normalised to the protein-coding gene *GAPDH* and shown as relative to DMSO, mean ± SEM, n = 4 biological replicates. Statistical test: two-tailed unpaired t-test. P-value: * < 0.05, ** < 0.01, *** < 0.001, **** < 0.0001. (**B**) Total pol II ChIP-qPCR on the intron-containing gene *KPNB1* in HeLa cells treated with DMSO or 30 μM Isoginkgetin for 1h to 6h. Mean ± SEM, n = 3 biological replicates. Statistical test: two-tailed unpaired t-test. P-value: * < 0.05, ** < 0.01. (**C**) Total pol II ChIP-qPCR on the intronless gene *JUN* in HeLa cells treated with DMSO or 30 μM Isoginkgetin for 1h to 6h. Mean ± SEM, n = 3 biological replicates. Statistical test: two-tailed unpaired t-test. P-value: * < 0.05. (**D**) Total pol II ChIP-qPCR of the histone gene *H1-2* in HeLa cells treated with DMSO or 30 μM Isoginkgetin for 1h to 6h. Mean ± SEM, n = 3 biological replicates. Statistical test: two-tailed unpaired t-test. P-value: * < 0.05. (**E**) Western blots of total pol II, Ser2-P, Ser5-P, and histone H3 on the chromatin fraction of HeLa cells treated

with DMSO or 30 μM Isoginkgetin for 1h, 2h, or 4h. Quantifications of the western blots are shown below each panel. Each quantification has been normalised to histone H3 signal and then to the DMSO condition. (**F**) ChIP-qPCR of total pol II, Ser2-P, and Ser5-P across the intron-containing gene *KPNB1* in HeLa cells treated with DMSO or 30 μM Isoginkgetin for 1h, 1h30, or 2h. Mean ± SEM, n = 3 biological replicates. Statistical test: two-tailed unpaired t-test. P-value: * < 0.05, ** < 0.01, *** < 0.001, **** < 0.0001.

of these genes. This upregulation may be partially explained by the transcriptional readthrough previously observed for Isoginkgetin [30, 40] (S4 Fig) from upstream protein-coding genes into ncRNA genes, transforming them into expressed genes in the differential expression analysis software. However, we could not observe a transcription termination defect nor the increase in total, but not poly(A)+, RNA-seq signal in the first intron [30] following Isoginkgetin treatment by total pol II ChIP-qPCR. As Isoginkgetin mimics RNA exosome inhibition [40], it is therefore likely that the accumulation of poorly processed transcripts (increased signal in the first intron, increased signal downstream of the poly(A) site, and increased non-polyadenylated ncRNAs) are due to a loss of RNA exosome activity rather than a transcriptional defect. The cellular target(s) of Isoginkgetin is also unknown. Isoginkgetin can partially inhibit SENP1 [45], a protease deconjugating SUMO modification, *in vitro*, and in agreement with this, we start to see an increase in global SUMO2/3 levels after 4h treatment. It remains to be determined whether the effect of Isoginkgetin on pol I and pol II transcription is mediated by rapid changes (< 4h) in SUMO modifications on proteins involved in transcription.

The effects of Isoginkgetin and Madrasin on transcription and pre-mRNA splicing also differ from the effects of SF3B1 inhibitors as we and others have found that PlaB rapidly inhibits co-transcriptional splicing (30 min) and reduces P-TEFb recruitment to chromatin, resulting in an increased pol II pausing, reduced transcription elongation, and increased premature transcription termination [5, 7, 19–21]. These results indicate that Isoginkgetin and Madrasin are unlikely to be SF3B1 inhibitors.

Understanding the coupling between transcription, pre-mRNA splicing, and mRNA CPA remains technically challenging. Based on our and other groups' analyses, Madrasin and Isoginkgetin should not be considered splicing inhibitors. It is also important to use short time-points to screen for potential splicing inhibitors rather than 24h treatment in order to limit the influence of secondary/indirect effects.

## Supporting information

**S1 Fig. Madrasin decreases transcription of protein-coding genes.** (**A**) RT-PCR with primers amplifying the intron located between exons 4 and 5 of *BRD2*. HeLa cells were treated with DMSO or 30 μM, 60 μM, or 90 μM Madrasin for 30 min. The location of the spliced and unspliced RNA is shown on the left of the panel. The percentages of unspliced RNA compared to total (spliced + unspliced) are shown below. (**B**) qRT-PCR with primers amplifying a region from the pol I transcribed pre-rRNA. HeLa cells were treated with DMSO or 90 μM Madrasin for 30 min (red) or 60 min (orange). cDNA was generated with random hexamers. Values are normalised to the 7SK snRNA and shown as relative to DMSO, mean ± SEM, n = 3 biological replicates. Statistical test: two-tailed unpaired t-test. P-value: n.s. not significant. (**C**) Metagene profile of pol II CTD Ser5-P mNET-seq performed in HeLa cells treated with DMSO (blue) or 90 μM Madrasin for 30 min (red) on scaled expressed protein-coding genes.
(TIF)

**S2 Fig. Transcription of intronless and histone genes is also affected by Madrasin.** (**A**) Violin plots, defined as min to max values, first and third quartiles (thin dotted lines), and mean (thicker dotted line), showing the $\log_{10}$ of gene length distribution (in bp) for intron-

containing (red), intronless (blue), and histone (black) genes. Statistical test: Wilcoxon rank sum test. P-value: *** < 0.001, **** < 0.0001. **(B)** Violin plots, defined as min to max values, first and third quartiles (thin dotted lines), and mean (thicker dotted line), showing the $\log_{10}$ of gene expression distribution, from two biological HeLa chromatin RNA-seq experiments, for intron-containing (red), intronless (blue), and histone (black) genes. Statistical test: Wilcoxon rank sum test. P-value: ns: not significant, **** < 0.0001. **(C)** Metagene profile of total pol II mNET-seq performed in HeLa cells treated with DMSO (blue) or 90 μM Madrasin for 30 min (red) on scaled expressed intron-containing genes. **(D)** Screenshots of the genome browser total pol II mNET-seq DMSO (blue) and Madrasin (red) tracks of the intronless genes *SOX4* and *SF3B5*. The arrow indicates the sense of transcription. **(E)** Ratios of SPT5 / total pol II, CDC73 / total pol II, or CPSF73 / total pol II from ChIP-qPCR on the intronless gene *JUN* in HeLa cells treated with DMSO (blue) or 90 μM Madrasin for 30 min (red). Mean ± SEM, n = 3 biological replicates. Statistical test: two-tailed unpaired t-test. P-value: * < 0.05. **(F)** Screenshot of the genome browser total pol II mNET-seq DMSO (blue) and Madrasin (red) tracks of the histone gene *H1-4*. The arrow indicates the sense of transcription. **(G)** Ratios of SPT5 / total pol II, CDC73 / total pol II, or CPSF73 / total pol II from ChIP-qPCR on the histone gene *H1-2* in HeLa cells treated with DMSO (blue) or 90 μM Madrasin for 30 min (red). Mean ± SEM, n = 3 biological replicates. Statistical test: two-tailed unpaired t-test. P-value: * < 0.05, *** < 0.001.
(TIF)

**S3 Fig. Isoginkgetin treatment affects pol I and pol II transcription. (A)** MA plot of total RNA-seq in HeLa cells treated with DMSO or Isoginkgetin for 6h or 18h. Significantly upregulated genes are shown in red while significantly downregulated genes are shown in blue. Other genes are shown in grey. The numbers of differentially expressed genes are shown on the figure. **(B)** MA plot of total RNA-seq in HeLa cells treated with DMSO or Isoginkgetin for 12h. Significantly upregulated genes are shown in red while significantly downregulated genes are shown in blue. Other genes are shown in grey. The numbers of differentially expressed genes are shown on the figure. **(C)** qRT-PCR with primers amplifying different protein-coding genes. HeLa cells were treated with DMSO or 30 μM Isoginkgetin for 24h. cDNA was generated with oligo(dT). Values are normalised to the protein-coding gene *GAPDH* and shown as relative to DMSO, mean ± SEM, n = 3 biological replicates. Statistical test: two-tailed unpaired t-test. P-value: * < 0.05, ** < 0.01, **** < 0.0001. **(D)** Total pol II ChIP-qPCR of the intron-containing gene *KPNB1* in HeLa cells treated with DMSO (black) or 30 μM Isoginkgetin (white) for 30 min. Mean ± SEM, n = 3 biological replicates. Statistical test: two-tailed unpaired t-test. P-value: not significant. **(E)** Western blot on whole cell extract of HeLa cells treated with DMSO or 30 μM Isoginkgetin for 1h, 2h, or 4h. The antibodies are indicated at the bottom, with Ponceau S staining used as loading control.
(TIF)

**S4 Fig. Isoginkgetin treatment affects transcription termination in RNA-seq. (A)** Metagene profile of total RNA-seq performed in HeLa cells treated with DMSO (blue) or Isoginkgetin (red) for 12h on scaled expressed intron-containing, intronless, and histone genes. The metagene profiles are shown as the average of two biological replicates. **(B)** Screenshots of the genome browser tracks for HeLa RNA-seq treated with DMSO (blue) or Isoginkgetin (red) for 12h on the intron-containing gene *KPNB1*, intronless genes *SOX4*, *SF3B5*, and *JUN*, and histone genes *H1-2*, *H1-4*, and *H2BC5*. The arrow indicates the sense of transcription.
(TIF)

## Acknowledgments

We thank the High-Throughput Genomics Group at the Wellcome Trust Centre for Human Genetics for sequencing.

## Author Contributions

**Conceptualization:** Michael Tellier, Shona Murphy.

**Data curation:** Michael Tellier, Gilbert Ansa.

**Formal analysis:** Michael Tellier, Gilbert Ansa.

**Funding acquisition:** Shona Murphy.

**Investigation:** Michael Tellier, Gilbert Ansa.

**Methodology:** Michael Tellier, Gilbert Ansa.

**Project administration:** Michael Tellier, Shona Murphy.

**Software:** Michael Tellier.

**Supervision:** Michael Tellier, Shona Murphy.

**Validation:** Michael Tellier, Gilbert Ansa.

**Visualization:** Michael Tellier, Gilbert Ansa.

**Writing – original draft:** Michael Tellier.

**Writing – review & editing:** Michael Tellier, Gilbert Ansa, Shona Murphy.

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
