## [Decision Letter · Decision Letter 0]

13 Aug 2024

PONE-D-24-29298Isoginkgetin and Madrasin are poor splicing inhibitorsPLOS ONE

Dear Dr.  Tellier,

Thank you for submitting your manuscript to PLOS ONE. After careful consideration, both the reviewers are enthusiastic about the report, however Reviewer 1 has some suggestions for minor revision.  Therefore, we invite you to submit a revised version of the manuscript that addresses the points raised during the review process.

We look forward to receiving your revised manuscript.

Kind regards,

Swati Palit Deb

Academic Editor

PLOS ONE

Journal Requirements:

   "This work was supported by the Wellcome Trust Investigator Awards [WT106134AIA and WT210641/Z/18/Z to S.M.]. Funding for open access charge: University of Leicester."

Reviewers' comments:

Reviewer's Responses to Questions

**Comments to the Author**

1. Is the manuscript technically sound, and do the data support the conclusions?

Reviewer #1: Yes

Reviewer #2: Yes

2. Has the statistical analysis been performed appropriately and rigorously? 

Reviewer #1: Yes

Reviewer #2: Yes

3. Have the authors made all data underlying the findings in their manuscript fully available?

Reviewer #1: Yes

Reviewer #2: Yes

4. Is the manuscript presented in an intelligible fashion and written in standard English?

Reviewer #1: Yes

Reviewer #2: Yes

5. Review Comments to the Author

Reviewer #1: 1. It would be great to show some of the qPCR data using the ddPCR method to understand the exact defect in the splicing or at the transcriptional level.

2. How are RNA Pol II subunit expression levels altered following Madrasin treatment compared to Isoginkgetin? Also, as a positive control, some of the figures in Figure 3 (such as 3F) are better suited for a PladB treatment comparison.

3. Similarly, for short-term treatment of Isoginkgetin, PladB comparison will be helpful to justify that splicing is secondary to transcriptional inhibition.

Reviewer #2: In this study, authors have investigated the effect of Madrasin and Isoginkgetin, two non-SF3B1 splicing inhibitors, on splicing and transcription. The results show that Madrasin and Isoginkgetin are poor splicing inhibitors using assays like RT-PCR for exons 2 and 3 of DNAJB1 and exons 4 and 5 of BRD2, nascent transcription activity via 5’EU incorporation, metagene profiling of total pol II mNET-seq and qRT-PCR with primers amplifying different protein-coding genes. My recommendation is to accept the manuscript in current without any additional changes.

6. PLOS authors have the option to publish the peer review history of their article (what does this mean?). If published, this will include your full peer review and any attached files.

Reviewer #1: No

Reviewer #2: No

---

## [Author Response · Author response to Decision Letter 0]

22 Aug 2024

Reviewer #1: 1. It would be great to show some of the qPCR data using the ddPCR method to understand the exact defect in the splicing or at the transcriptional level.

We thank the reviewer for their positive comments. While interesting, we think that ddPCR is beyond the scope of the study as the main message of the manuscript is that Isoginkgetin and Madrasin are poor splicing inhibitors and should therefore not be used to inhibit pre-mRNA splicing.

2. How are RNA Pol II subunit expression levels altered following Madrasin treatment compared to Isoginkgetin? Also, as a positive control, some of the figures in Figure 3 (such as 3F) are better suited for a PladB treatment comparison. 3. Similarly, for short-term treatment of Isoginkgetin, PladB comparison will be helpful to justify that splicing is secondary to transcriptional inhibition.

We added to Fig 1E western blots of the large subunit RNA polymerase II (Rpb1), SF3B1, α-tubulin, and histone H3 performed on chromatin/nucleoplasm/cytoplasm fractions of HeLa cells treated for 1 hour with DMSO, Isoginkgetin, Madrasin, Pladienolide B, and Herboxidiene. While SF3B1 (as positive control) is decreased on chromatin and increased in nucleoplasm and cytoplasm fractions after treatment with PlaB, HB, and Madrasin, Isoginkgetin has no effect on SF3B1. For total Rpb1, none of the inhibitors has an effect on its cellular distribution indicating that the decrease in transcription is not due to degradation of pol II.

For the comparison of Isoginkgetin and Madrasin to PlaB, we have extended the Discussion to describe the results we obtained on the effect of PlaB on splicing and transcription and how they differ to what we observe for Isoginkgetin and Madrasin. 

Added to the Discussion- “The effects of Isoginkgetin and Madrasin on transcription and pre-mRNA splicing also differ from the effects of SF3B1 inhibitors as we and others have found that PlaB rapidly inhibits co-transcriptional splicing (30 min) and reduces P-TEFb recruitment to chromatin, resulting in an increased pol II pausing, reduced transcription elongation, and increased premature transcription termination (5, 7, 19-21). These results indicate that Isoginkgetin and Madrasin are unlikely to be SF3B1 inhibitors.”

Reviewer #2: In this study, authors have investigated the effect of Madrasin and Isoginkgetin, two non-SF3B1 splicing inhibitors, on splicing and transcription. The results show that Madrasin and Isoginkgetin are poor splicing inhibitors using assays like RT-PCR for exons 2 and 3 of DNAJB1 and exons 4 and 5 of BRD2, nascent transcription activity via 5’EU incorporation, metagene profiling of total pol II mNET-seq and qRT-PCR with primers amplifying different protein-coding genes. My recommendation is to accept the manuscript in current without any additional changes.

We thank the reviewer for their positive comments.

---

## [Editor Report · Decision Letter 1]

3 Sep 2024

Isoginkgetin and Madrasin are poor splicing inhibitors

PONE-D-24-29298R1

Dear Dr. Tellier,

We’re pleased to inform you that your manuscript has been judged scientifically suitable for publication and will be formally accepted for publication once it meets all outstanding technical requirements.

Within one week, you’ll receive an e-mail detailing the required amendments. When these have been addressed, you’ll receive a formal acceptance letter, and your manuscript will be scheduled for publication.

An invoice will be generated when your article is formally accepted. Please note, if your institution has a publishing partnership with PLOS and your article meets the relevant criteria, all or part of your publication costs will be covered. Please make sure your user information is up to date by logging into Editorial Manager at Editorial Manager® and clicking the ‘Update My Information' link at the top of the page. If you have any questions relating to publication charges, please contact our Author Billing department directly at authorbilling@plos.org.

Kind regards,

Swati Palit Deb

Academic Editor

PLOS ONE
---

## [Editor Report · Acceptance letter]

10 Oct 2024

PONE-D-24-29298R1 

PLOS ONE

Dear Dr. Tellier, 

I'm pleased to inform you that your manuscript has been deemed suitable for publication in PLOS ONE. Congratulations! Your manuscript is now being handed over to our production team.

Kind regards, 

on behalf of

Dr. Swati Palit Deb 

Academic Editor

PLOS ONE